# Improved Weathering Performance of Poly(Lactic Acid) through Carbon Nanotubes Addition: Thermal, Microstructural, and Nanomechanical Analyses

**DOI:** 10.3390/biomimetics5040061

**Published:** 2020-11-17

**Authors:** Thevu Vu, Peyman Nikaeen, William Chirdon, Ahmed Khattab, Dilip Depan

**Affiliations:** 1Institute of Materials Research and Innovation, Department of Chemical Engineering, University of Louisiana at Lafayette, P.O. Box 43675, Lafayette, LA 70504-4130, USA; vu.vu1@louisiana.edu (T.V.); william.chirdon@louisiana.edu (W.C.); 2Laboratory of Composite Materials, Department of Mechanical Engineering, University of Louisiana at Lafayette, P.O. Box 43675, Lafayette, LA 70504-4130, USA; peyman.nikaeen1@louisiana.edu (P.N.); ahmed.khattab@louisiana.edu (A.K.)

**Keywords:** carbon nanotubes, poly(lactic acid), crystallization, microstructure, accelerated weathering, degradation, nanoindentation

## Abstract

To understand the interrelationship between the microstructure and degradation behavior of poly(lactic acid) (PLA), single-walled carbon nanotubes (CNTs) were introduced into PLA as nucleating agents. The degradation behavior of PLA-CNT nanocomposites was examined under accelerated weathering conditions with exposure to UV light, heat, and moisture. The degradation mechanism proceeded via the Norrish type II mechanism of carbonyl polyester. Differential scanning calorimetry (DSC) studies showed an increase in glass transition temperature, melting temperature, and crystallinity as a result of the degradation. However, pure PLA showed higher degradation as evidenced by increased crystallinity, lower onset decomposition temperature, embrittlement, and a higher number of micro-voids which became broader and deeper during degradation. In the PLA-CNT nanocomposites, CNTs created a tortuous pathway which inhibits the penetration of water molecules deeper into the polymer matrix, making PLA thermally stable by increasing the initial temperature of mass loss. CNTs appear to retard PLA degradation by impeding mass transfer. Our study will facilitate designing environmentally friendly packaging materials that display greater resistance to degradation in the presence of moisture and UV light.

## 1. Introduction

Poly(lactic acid) (PLA), an aliphatic polyester, is one of the most promising biopolymers to replace conventional petroleum-based plastics [1]. The use of biopolymers has attracted significant attention because of increasing environmental concerns due to the waste accumulation and depletion of fossil resources [2,3,4]. PLA is a sustainably sourced material and can be derived from renewable resources such as corn, rice, or wheat. With the advantages of being a bio-compatible, hydrolysable, and biodegradable material, PLA has great potential for both packaging and biomedical applications [1,5]. However, PLA has some limitations that prevent it from becoming a widely used material including its brittleness as well as its low heat and lower water resistance [6,7]. To address these limitations, PLA has been mixed with nanofillers to create new nanocomposite materials. Nanocomposites can yield significantly improved properties at very low filler contents [8].

Among the commonly used nanoparticles, carbon nanotubes (CNTs) have been widely exploited in research due to their excellent physical and mechanical properties [8,9,10]. It is reported that the presence of CNTs has a dual effect: nucleating both melt crystallization and cold crystallization, but also impeding the crystal growth by acting as a physical barrier [11,12,13,14]. In fact, the presence of CNTs often leads to the formation of smaller crystalline structures and prevents the formation of larger spherulites [13].

Since PLA has been used in packaging applications, the degradation rate of this thermoplastic is a key consideration which determines the longevity of the material and is strongly affected by environmental conditions. Exposure to sunlight is known to cause degradation, discoloration and brittle fracture [8,15]. This complex process follows hydrolysis or oxidation, resulting into the breakage of polymer chain scission and breakdown of the polymer crystal structure. When solar radiation strikes the polymer surface, photons with energy similar or higher than the chemical bond strength of the polymer causes a series of reactions that can lead to polymer chain scission and eventual deterioration of polymer properties [9]. On the other hand, in the presence of moisture, hydrolytic degradation will occur, where water molecules diffuse into the polymer and cause ester cleavage and main-chain scission [16]. It is reported that degradation primarily occurs in the amorphous region of the polymer [12,17], hence polymers with higher crystallinity are more resistant to degradation. There has been a substantial amount of work that has been addressed on the degradation of PLA, including hydrolytic degradation [16], photodegradation [15,18], and composting conditions [19]; however, these degradation conditions were studied independently. Only few studies reported the effect of a combination of ultraviolet (UV) irradiance, heat, and moisture, which closely describes the actual weathering environment [17,20,21]. Furthermore, to the best of authors‘ knowledge, investigation of the weathering performance of PLA-CNTs nanocomposites under these accelerated weathering conditions has not yet been reported. Testing PLA under such conditions would provide more valuable information for outdoor applications and waste disposal. Moreover, CNTs modulate the structure and size of PLA crystals. Since the observable macroscopic degradation behavior is controlled by the degradation on a microstructural scale, it is expected that morphological changes caused by CNTs would affect the degradation of PLA-CNT nanocomposites. Yet, there is no work reported on the influence of CNTs on PLA weathering resistance. Thereby, it is essential to understand how the change in structures obtained by CNTs affect the photolysis and hydrolysis degradation of PLA.

Present work consists of two parts. In the first part, the influence of CNTs on PLA isothermal crystallization was investigated. Changes in microstructures and thermal properties between PLA and PLA-CNT nanocomposites were evaluated. In the second part, PLA and PLA-CNT nanocomposites films were subjected to a combination of UV irradiance, heat and moisture. Comparison of the degradation rates between PLA and PLA-CNT nanocomposites was quantified by monitoring the crystallinity, thermal stability, and nanomechanical properties at different exposure times. The research presented here provides a deeper understanding of the photo-hydrolytic degradation of PLA and PLA-CNT nanocomposites by investigating how different microstructures affect the degradation rate.

## 2. Materials and Methods

### 2.1. Materials

Poly (L-lactic acid) (PLLA, 4032D) with a length to diameter ratio of 24:1, a melting point of 180 °C, glass transition temperature of 55 °C–60 °C, melt flow index range at 190 °C under 2.16 kg of 2–8 g/10 min, and a density of 1.25 g/cm^3^ was supplied by Natureworks (Minnetonka, MN, USA). Single wall carbon nanotubes with a 2 nm outer diameter, a length of 5–30 µm, and 95% purity. were obtained from Nanostructured and Amorphous Materials (Katy, TX, USA), Tetrahydrofuran (THF) with less than 20 ppm peroxide and less than 0.002% water impurity was obtained from Sigma Aldrich (Saint Louis, MO, USA).

### 2.2. Thin Film Fabrication

Pure PLA and PLA-CNT 1 wt.% nanocomposites were prepared as thin films in three steps: solution mixing, melt-mixing, and compression molding using a Carver hot press (Figure 1). First, PLA pellets were pre-dried in a vacuum oven at 40 °C for at least 4 h. Then, 2 g of PLA was dissolved in 15 mL of THF and stirred for 1 h at 50 °C. Next, 0.02 g of CNTs were dispersed in 15 mL of THF by ultra-sonicating for 1 h. After sonication, the CNT-THF dispersion was added into PLA solution. The mixture was then continuously stirred at 50 °C for two hours. Next, the resulting mixture was poured onto a glass petri dish and then dried at room temperature. The dried PLA-CNT mixture was then mixed again using a fully automated laboratory Brabender mixer (Brabender, Duisburg, Germany). The temperature was set at 200 °C, and the rotation speed was set as 30 rpm for 3 min and then 70 rpm for 7 min. This mixture of well-dispersed CNT in PLA was transferred to a compression molding station. Two Carver press plates were used: one heated at 180 °C for 10 min to melt and remove thermal history and the other was heated at 110 °C for 60 min for crystallization. After crystallization, thin films of PLA-CNT nanocomposite were obtained with a thicknesses range of 0.3–0.4 mm.

### 2.3. Accelerated Weathering Test

A UV Accelerated Weathering Tester (Q-Panel Q-Lab Westlake, OH, USA) was used to conduct the accelerated weathering tests. Testing conditions were in accordance with cycle 1 of ASTM 154-06, SAEJ2020, and ISO48892-3 [21]. Fluorescent lamps (UVA-340) with 0.89 W/m^2^ (at 340 nm) irradiance were used with cycles of 8 h UV exposure irradiation at 60 °C followed by 4 h dark condensation at 50 °C. Tap water was used as the water feed. The reason for light off during wet interval is to simulate the dew effect because dew plays a key role in determining material durability [22]. The accelerated test was performed for three periods: 100 h, 200 h, and 300 h. After each period, specimens were taken out for further analysis.

### 2.4. Scanning Elecron Microscopy (SEM)

Differences in the surface morphology between the neat PLA and the PLA-CNT 1 wt.% specimens were observed under a 6300 scanning electron microscope (SEM, JEOL, Peabody, MA, USA). All specimens used for SEM analysis were etched with a solution of methanol, water, and sodium hydroxide, and then vacuum dried overnight at 40 °C before being coated with sputtered gold layer of 18 nm. Image-J 1.8 software (NIH, Madison, WI, USA) was used to conduct a morphometric analysis from SEM images to determine the number of voids and observed surface porosity. Selected SEM images were transformed to binary 8-bit images and thresholds were automatically applied using the built-in “intermodes” method. Then, the particle analyzer toolbox was used to count the number of voids on the binary images. Minimum size of void was set at 10 pixels on 535 × 435 resolution images to avoid counting the noise existing in the images.

### 2.5. Thermal Analysis

Thermal analysis was performed using a DSC400 instrument (Perkin Elmer, Waltham, MA, USA) under a nitrogen atmosphere. A small (~10 mg) sample was heated from 30 °C to 200 °C with a heating rate of 10 °C per minute. Thermal stability was assessed with a thermogravimetric analysis (TGA 100, Instrument Specialists Incorporated, Boerne, TX, USA). Samples were heated in open platinum pan from 25 °C to 500 °C, under nitrogen atmosphere to avoid thermoxidative degradation, at a heating rate of 10 °C/min. The degree of crystallinity, *X_c_* of samples was determined by the following expression:$$X_{c}\left(\%\right)=\frac{\Delta H_{f}-\Delta H_{c}}{\Delta H_{f}^{0}}\;\times 100$$
where $\Delta H_{f}$ is the melting enthalpy and $\Delta H_{f}^{0}$ is the melting enthalpy of pure crystalline PLA (100% crystalline polylactide = 93 J/g) and $\Delta H_{c}$ is the crystallization enthalpy [23].

### 2.6. Nanoindentation

Nanoindentation is a non-destructive tool for testing thin films and surface mechanical properties of a wide range of materials including polymers [24,25,26,27]. In this study, nanoindentation technique was employed to measure micromechanical properties of the PLA/PLA-CNT thin films before and after degradation. Indentation tests were performed at room temperature utilizing an XP G200 from Agilent Instruments (Santa Clara, CA, USA) equipped with a three-sided Berkovich pyramidal diamond tip by continuous stiffness measurement (CSM) method. CSM uses a superimposed sinusoidal load on top of the main motion driving force to record the elastic modulus and hardness of the material as a function of penetration depth [28]. Properties were measured up to the maximum depth of 10 µm which is in an acceptable range to avoid any substrate effect considering the thickness of the thin films (0.25–0.3 mm) [29]. The harmonic displacement of the sinusoidal load was set as 2 nm with a 45 Hz frequency and the strain rate was chosen as 0.05 s^−1^. Prior to the actual tests on the thin films, tip area calibration was performed by conducting 25 indentations on a reference fused silica sample (with E = 72 GPa). A typical load-depth response of the tests is shown in Figure 2. Since the mechanical behavior of the polymers has shown to be strain rate-dependent, properties are measured at constant strain rate to avoid any discrepancies between different samples (segment 1 of the curve shown in Figure 2). This allows for an appropriate statistical analysis and comparison between the data extracted from the samples undergone different degradation conditions. Segment 2 represents the hold segment between the loading and unloading stages which is to eliminate the effect of creep behavior of the polymeric materials in the measurement procedure [30]. The analysis of the load-displacement curve to calculate the properties was carried out based on the method developed by Oliver and Pharr [31]. For each sample, 25 indentations were conducted in a grid of 5 × 5 points having a 250 µm distance between adjacent indents to avoid any interactions between their plastic regions. 25 data points for each sample/condition were statistically analyzed and the mean and standard deviation were used in the discussion.

## 3. Results and Discussion

### 3.1. CNT-Induced PLA Crystallization

#### 3.1.1. Morphological Studies

The PLA and PLA-CNT nanocomposites were etched to reveal the microstructure before examination by SEM. The resultant micrographs are presented in Figure 3. After etching, pure PLA revealed large spherulites as indicated by circles in Figure 3a.

A closer observation of these spherulites revealed small centers in the middle with long lamellar fibrils growing outward. As shown in Figure 3b, the average diameter of these spherulites is 20 µm with well-developed, round shape since their growth was not impeded by other spherulites or nanoparticles. Adjacent spherulites were separated by amorphous regions with an average length of 4 to 6 µm. The micrographs of PLA-CNT exhibited a much higher (number) density of spherulites due to the presence of CNTs (Figure 3c,d). As reported previously [11,12,13,14,32], CNTs can act as heterogeneous nucleating agents to provide surface area for the polymer to crystallize. Therefore, the presence of CNTs increased the nucleation density which prevents large spherulites from forming, resulting in the formation of smaller and less organized spherulites. The average radius of PLA-CNT spherulites was only ~5 µm.

#### 3.1.2. Thermal Behavior

The first heating thermograms (Figure 4) of PLA and PLA-CNT after isothermal crystallization indicated the absence of cold crystallization as there is no exothermic peak found on these curves, suggesting that most of the crystallizable polymer was incorporated into crystalline structures during the annealing at 110 °C. On the pure PLA thermogram, there was a small endothermic transition with a peak temperature of 69.63 °C rather than the common step change indicative of the glass transition temperature (Tg). This is a known phenomenon that has been reported to be the result of relaxation of the amorphous phase [17,33]. However, this relaxation did not appear on the thermogram of PLA-CNT due to the low content of amorphous phase which is constrained by the predominant crystalline phase [33]. As shown in Table 1, pure PLA and PLA-CNT exhibit the Tg values of 61.27 °C and 61.87 °C, respectively. This negligible change in the glass transition temperature between pure PLA and PLA-CNT was observed is likely due to the low content of nanofillers [34]. Since the CNTs loading was low, only a small reduction in PLA chain mobility was caused by the nanofillers.

Another observation is the bimodality of fusion peaks on both DSC curves. For PLA, the highly ordered α-crystal is known to be the most frequently observed structure when crystallizing above 120 °C, while the relatively disordered α’-crystal is observed when crystallizing below 100 °C [35,36]. If the crystallization temperature is between 100 °C and 120 °C, both forms crystallize, and α’ will spontaneously transform into the more stable α phase during heating. Hence, the double melting peak observed is an indication of the existence of both α and α’ crystalline phases in PLA with and without CNTs. This is in agreement with previous findings that CNTs have no effect on the crystalline form of PLA [16]. However, the peak temperature and magnitude of these two crystalline structures slightly changed after adding CNTs. Pure PLA crystals started to melt at 159.6 °C, and the first peak was observed at 166.9 °C, while majority of crystallites was formed at the second peak at 173.6 °C. With the reinforcement of CNT, onset melting temperature increased to 160.7 °C, while the peak melting temperatures of bimodal peaks changed to 168.0 °C and 172.8 °C, respectively. Interestingly, the magnitude of first peak increased significantly with the inclusion of CNTs which indicates that PLA was crystallizing at much lower crystallization temperature. It happened due to the presence of CNTs where they acted as nucleating agents to accelerate overall crystallization [29]. Similar results have been observed in the past study [37]. Hence, the PLA-CNT crystals melt at slightly higher temperature. Lastly, in PLA-CNT, there was a small endothermic peak around 143.3 °C due to a small amount of mesophase transitioning to an isotropic liquid [15]. This mesophase exists as a “transition layer” or “interface” between the crystalline and amorphous regions of semicrystalline polymers [38].

### 3.2. Photo-Hydrolytic Degradation

#### 3.2.1. Thermal Transition during Degradation

Thermal analysis of the pure PLA and PLA-CNT nanocomposite was conducted using DSC after the weathering tests. As shown in Table 1, the Tg of pure PLA was increased from 61.3 °C to 63.4 °C. Meanwhile, Tg of PLA-CNT increased from 61.8 °C to 64 °C after 100 h of degradation. The tendency of increasing Tg at the beginning of the degradation process was previously reported [39]. This is likely due to stabilized chain packing in the amorphous region by low temperature annealing in the presence of water molecules which are acting as a plasticizer [39]. Hence, more energy is required to activate the glass-rubber transition, resulting in an increase in Tg. This increase in Tg leads to embrittlement which is one of the deleterious effects of degradation. However, as the degradation proceeded, Tg gradually decreased. Tg of pure PLA decreased from 63.4 °C to 61.3 °C after 200 h and 60.3 °C after 300 h of degradation. On the other hand, Tg of PLA-CNT decreased from 64 °C to 60.5 °C after 200 h, and 59.2 °C after 300 h of degradation. This is expected since polymer chains started to break into shorter chains and these smaller chains requires less energy to become mobile. In addition, water molecules could be dissolving into PLA to increase the free volume.

Figure 5a shows DSC heating curves of pure PLA after 100 h, 200 h, and 300 h of accelerated weathering. It can be seen that the temperatures of the double melting peak of pure PLA gradually increased during degradation. The melting temperature went from 166.9 °C and 173.61 °C at 0 h to 169.08 °C and 173.91 °C after 300 h. The crystallinity increased monotonically from 38.9% at 0 h to 42.35% at 100 h, 43.6% at 200 h, and 45.15% at 300 h (Table 1). Similar result was found in Tsuji et al. where the relative crystallinity of amorphous PLA film increased rapidly in the first 12 h due to the crystallization on immersion in phosphate-buffered solution [40]. The increase in crystallinity with degradation is a well-known phenomenon: during degradation, chain scission occurs providing the polymer chains with extra mobility to rearrange into larger crystal structures [41]. Also, secondary crystallization can proceed below Tg over the time scale of months and years, so the PLA may have simply continued to crystallize over the extended period of time [42].

Figure 5b shows thermograms of PLA-CNT for various degradation times. The small endothermic peak that appeared at 143.26 °C in the original sample, which was identified as a small amount of mesophase [43], shifted to the right at 157.8 °C after 100 h and eventually merged into the double melting peaks. This is likely due to the fact that this intermediate state was less ordered than perfect crystals and started rearranging into more ordered crystals during degradation.

As seen in the results for the neat PLA, degradation showed its effect on the onset melting temperatures which decreased from 161.27 °C before degradation to 160.34 °C after 300 h of degradation. Meanwhile, onset melting temperatures of PLA-CNT nanocomposite remained unchanged throughout the degradation process. The temperatures of the double melting peaks of PLA-CNT increased as the degradation proceeded from 167.97 °C and 172.84 °C at 0 h to 171.73 °C and 176.31 °C after 300 h.

When compared to pure PLA, the higher onset temperature and peak temperatures indicate higher thermal stability of PLA-CNT after degradation. Further, the percent crystallinity of PLA-CNT increased from 41.4% at 0 h to 43.9% after 100 h but did not significantly increased beyond that point. This may also be attributed to secondary crystallization where the polymer continued to crystallize over a longer period of time but slowed as it approached equilibrium. Figure 6 shows the decomposition temperature of both pure PLA and PLA-CNT at different irradiation times.

The main degradation behavior of PLA and PLA-CNT nanocomposites were similar. The decomposition starts at around 300 °C, followed by a rapid weight loss of over 96% around 410–420 °C. Thereby, the onset decomposition temperature (ODT) was paid close attention since it is dependent on the molecular weight and crystallinity of polymer [44]. As expected, ODT slowly decreased as the degradation was prolonged due to the effect of chain scissions and molecular weight reduction. For PLA, ODT significantly decreased from 384 °C at 0 h of degradation to 369 °C, 352.5 °C, and 337 °C at 100 h, 200 h, and 300 h, respectively. On the other hand, in the case of PLA-CNT, ODT increased from 358.8 °C to 360.3 °C in the first 100 h of degradation, which is likely due to the increase in molecular weight by adding CNTs. At 200 h and 300 h, ODT decreased to 348.5 °C and 355.1 °C, respectively. While comparing between PLA and PLA-CNT, the composites exhibited higher thermal stability at all three degradation time periods. This is an indication of the higher molecular weight remained during degradation which could simply be a result of CNTs inclusion. However, without CNTs, pure PLA has shown a higher reduction rate for decomposition temperature which indicates that pure PLA rapidly lost its thermal stability during the degradation. In fact, after 300 h of degradation, ODT of PLA reduced to 12.2% while it was only 1% for PLA-CNT nanocomposites. This big difference in reduction of ODT after exposure to weathering condition for 300 h, strongly implies a lower degradation rate for the PLA-CNT nanocomposites which is mainly ascribed to the influence of the carbon nanotubes.

#### 3.2.2. Crystal Microstructure and Degradation: Direct Observation from SEM

Scanning electron microscopy revealed the morphology of PLA and PLA-CNT films after photo-hydrolytic degradation. As shown in Figure 7, PLA and PLA-CNT both showed the presence of voids after 300 h of degradation with an increase in the number of voids over time. Morphometric analysis was performed on the SEM images shown in Figure 7 to determine the number of voids, their sizes and porosity with the results presented in Table 2. On the PLA-CNT film, only one additional void appeared over a 0.018 mm2 area after the first 100 h of degradation while 33 voids were formed over the same size area of the pure PLA film. After 300 h of degradation, the number of voids found on pure PLA was four times higher than the number found on PLA-CNT. These number of voids is an indication of slower degradation rate of PLA-CNT nanocomposite compared to pure PLA film.

As discussed, void formation due to UV and moisture exposure resulted in an increasing number of voids on the surface of the polymer films over time. After 300 h, a significantly higher number of voids appeared on the pure PLA surface compared to the PLA-CNT nanocomposite. Ester bond cleavage is mainly responsible for this degradation which has been discussed in literature [21,42,45]. This ester bond cleavage can start from the surface which is directly exposed during weathering tests during UV and hydrolytic degradation. Expectedly, these micro voids first appeared in the amorphous regions in the polymer matrix since it has lower resistance to degradation than the one of crystalline regions. In fact, it has been reported that both photolysis and hydrolytic degradation would start from the amorphous regions first [16,40]. Tsuji et al. confirmed that the crystalline region has higher photodegradation resist than amorphous region while monitoring molecular weight distribution of UV treated and untreated samples [40]. In another study, it was reported that the chain cleavage reaction during the hydrolytic degradation of PLA would proceed preferentially in amorphous regions [16].

Further analysis was made by evaluating the microstructure of the voids at higher magnification. Figure 8 illustrates the differences in microstructure between pure PLA and PLA-CNT and shows the role of CNTs during PLA degradation. Pure PLA crystallizing as large spherulites is a well-known phenomenon. Since CNTs induced the crystallization of PLA, more spherulites were formed but with smaller sizes. However, during degradation, these PLA-CNT spherulites exhibited higher thermal stability. In the early degradation stages, voids formed in the amorphous phase, and as the degradation progressed, the number of voids increased while the existing voids broadened and penetrated deeper into the pure PLA matrix. Consequently, the broadening of these voids caused the film to become more porous, allowing for easier penetration of water molecules and easier escape for monomer degradation products (e.g., lactic acid) from the matrix.

In the case of PLA-CNT, CNTs are known to absorb UV light which reduces the depth of photolysis [18]. Moreover, the CNTs can physically arrange into a mesh-like barrier or web-like structure that creates a more tortuous path which inhibits the diffusion of water molecules and free radicals formed during photo-degradation into the depth of the film [41]. This inhibiting effect of nanomaterials was also found in PLA-clay. It was reported that by adding montmorillonite (Mt) in PLA, exfoliated Mt layers not only exhibited nanoscale reinforcing effect, but also showed barrier actions decreasing the detrimental effects of photo and hydrolytic degradation. In this study, CNTs are thought to exhibit an effect in PLA similar to nanoclays. Slower void formation in the PLA-CNT nanocomposite compared to pure PLA indicates slower degradation due to the effect of tortuous path created by the CNTs, limiting mass transfer. This attribute is important when PLA-CNT is used in tissue engineering applications, primarily as wound sutures. It is expected that the suture would keep the wound closure tightly at the early stage, exhibiting minimum or controlled degradation, in-vivo. However, as the wound heals gradually, the biodegradable suture is no longer needed and thus should degrade, with subsequent bio-resorption.

#### 3.2.3. Micromechanical Properties by Nanoindentation

Elastic modulus (E) and hardness (H) of the samples were recorded by CSM method throughout the indention loading segment. Figure 9 shows a typical graph of measured properties as a function of penetration depth up to 10 µm. To avoid the indentation size effect (ISE) which is observed for displacements below 2 μm, the CSM results were first scrutinized to find an optimum penetration depth beyond which the obtained measured properties are stable. Hence, according to the graph shown in Figure 9, all the CSM data are processed hereafter in the range of 4–10 μm.

Figure 10 presents the measured elastic modulus and hardness of all samples. The results at 0 h evidently show that the inclusion of 1% wt. CNT to PLA increased its modulus and hardness by 4.6% and 10%, respectively. This enhancement in mechanical properties is attributed to the reinforcing effect of CNTs that has been widely reported for polymeric materials [46,47,48].

The influence of degradation on mechanical properties was investigated by monitoring elastic modulus and hardness of PLA and PLA-CNT thin films over 300 h degradation. Results revealed that there were two distinct trends of the mechanical properties evolution for pure PLA and PLA-CNT nanocomposites throughout the degradation period. For the pure PLAs, both E and H gradually increased throughout the degradation time. Two-tailed t-test was performed on pairs of results as follow: (1) PLA samples at different degradation hours, (2) PLA-CNT samples at different degradation hours and (3) PLA and PLA-CNT samples at different degradation hours. Obtaining α values for the two-tallied t-tests revealed that the mechanical properties of pure PLA thin films significantly differ from one degradation hour level to another in contrast to the PLA-CNT samples where changes in mechanical properties were not statistically significant. PLA-CNT samples maintained their elastic modulus and hardness values around 5.6 GPa and 0.35 GPa, respectively throughout the whole degradation period.

The significant increase in properties of pure PLA sample can be explained by the increase of crystallinity as shown in DSC results (Table 1). Increase in crystallinity during degradation is thought to be the result of chemi-crystallization which is caused by chain scission [49]. Consequently, this chemi-crystallization would lead to the embrittlement of the material. The degradation-induced embrittlement mechanism can be explained as follows: chain scission → molecular weight (MW) decrease → chemi-crystallization → decrease of the interlamellar spacing → embrittlement [46]. Therefore, increase in modulus of pure PLA clearly indicates that the material became brittle during degradation which denotes the effect of chain scission. Meanwhile, CNTs act as stabilizers that help to delay the loss in MW, caused by degradation and retain the initial mechanical properties of the PLA-CNT samples as was confirmed by the nanoindentation results. Additionally, the increase in modulus and hardness of PLA films during degradation indicates that the effects of pore formation in the surface is slight compared to the increased crystallinity.

## 4. Conclusions

CNTs increased the crystallinity of the PLA and modified the resultant morphology. Both pure PLA and PLA-CNT crystallized under α and α’ forms. However, SEM showed pure PLA crystallized as large, well-developed spherulites, while PLA-CNT nanocomposites displayed higher spherulite densities with smaller, more irregular spherulites. CNTs act as nucleating agent, increasing the crystallinity of PLA; however, did not impact the glass transition temperature (Tg).At the beginning of the weathering test, the Tg increased, but as the degradation proceeded, Tg decreased, most likely due to the reduction of molecular weight from chain scission. The degree of crystallinity of pure PLA was found to significantly increase during the degradation. The spherulites formed in the PLA-CNT nanocomposites were slightly more thermally stable as indicated by the onset melting temperature.Up to 300 h of degradation, both modulus (E) and hardness (H) of pure PLA gradually increased. It indicates that the material became brittle which is due to the chain scissions and chemi-crystallization caused by degradation. On the other hand, E and H of PLA-CNT did not significantly change which is contributed to the influence of CNTs which act as a stabilizer and retain initial nanomechanical properties of PLA-CNT nanocomposites samples.The PLA-CNT nanocomposites degraded slowly compared to the pure PLA as observed by the slower void formation rate, lower decomposition temperature reduction, and the preservation in mechanical properties. Morphometric analysis showed fewer and smaller voids on the PLA-CNT nanocomposite film which would limit the penetration of water molecules into the depth of the film.

## Figures and Tables

**Figure 1 biomimetics-05-00061-f001:**
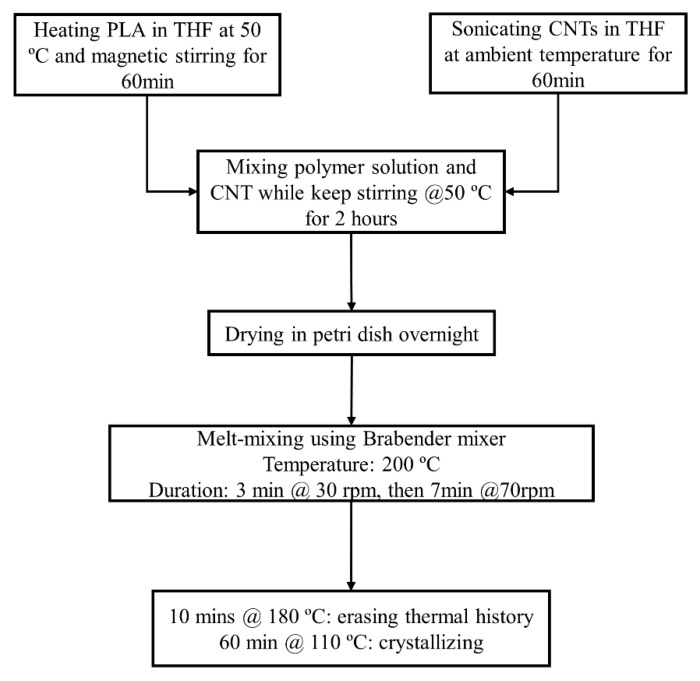
Fabrication process of the CNT-PLA thin films.

**Figure 2 biomimetics-05-00061-f002:**
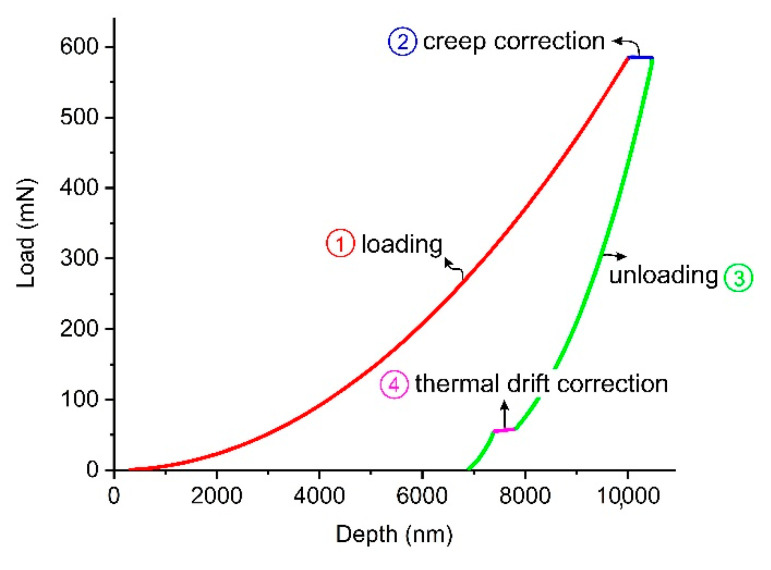
Typical load-displacement curve for the nanoindentation tests.

**Figure 3 biomimetics-05-00061-f003:**
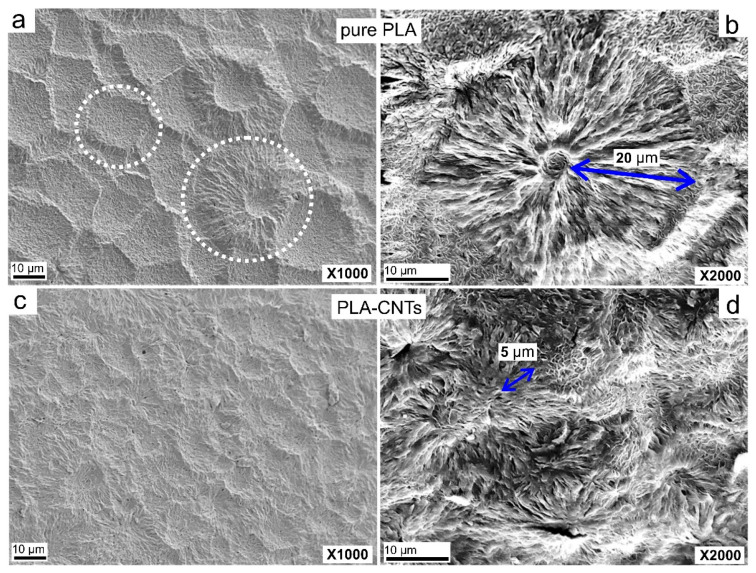
Scanning electron micrographs at 1000× and 2000× magnification of pure PLA (**a**,**b**), and PLA-CNT (**c**,**d**) after isothermal crystallization.

**Figure 4 biomimetics-05-00061-f004:**
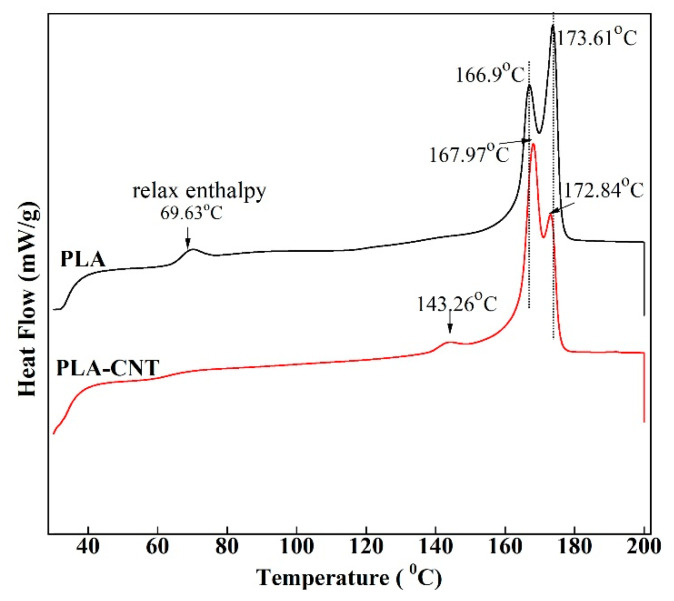
First heating differential scanning calorimetry (DSC) thermograms of PLA and PLA-CNT after isothermal crystallization.

**Figure 5 biomimetics-05-00061-f005:**
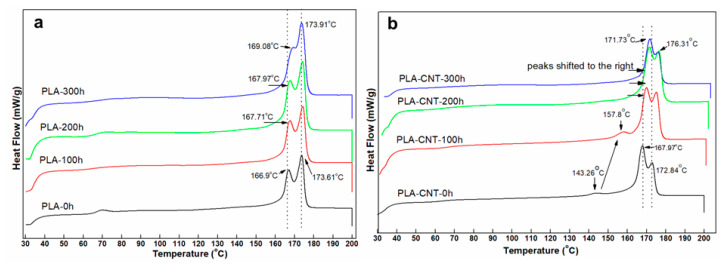
DSC of PLA (**a**) and PLA-CNT (**b**) after degradation.

**Figure 6 biomimetics-05-00061-f006:**
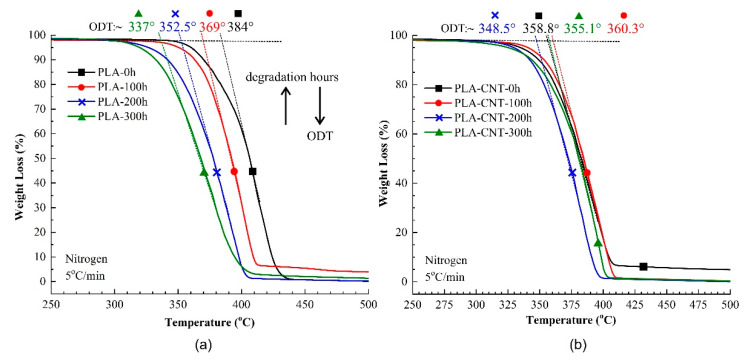
Comparison of thermogravimetric analysis (TGA) of (**a**) pure PLA and (**b**) 1% wt. PLA-CNT nanocomposites, at different degradation times.

**Figure 7 biomimetics-05-00061-f007:**
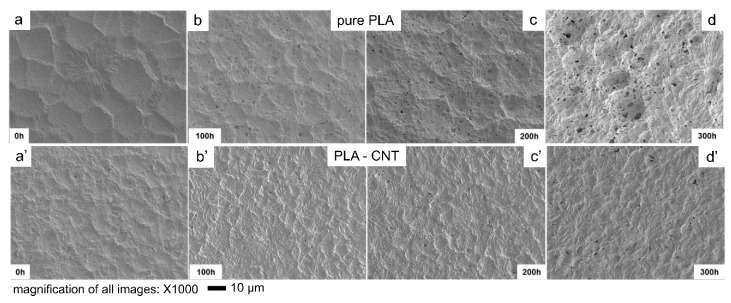
SEM results showing the surface morphology at 1000× magnification of (**a**–**d**) pure PLA, and (**a’**–**d’**) PLA-CNT after 300 h of degradation.

**Figure 8 biomimetics-05-00061-f008:**
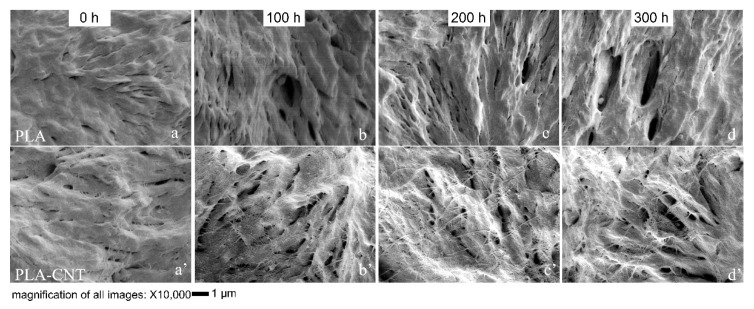
SEM images at 10,000× magnification illustrating the changes in the surface morphology in terms of voids and holes during 300 h of degradation; (**a**–**d**) PLA spherulic lamellae; (**a**’–**d**’) PLA-CNT spherulic lamellae.

**Figure 9 biomimetics-05-00061-f009:**
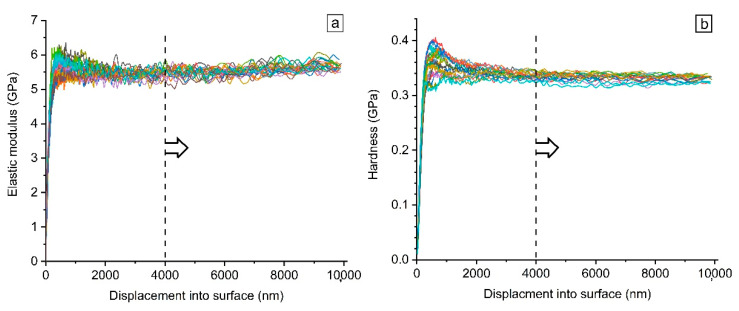
Typical variation of the (**a**) elastic modulus and (**b**) hardness with indentation depth.

**Figure 10 biomimetics-05-00061-f010:**
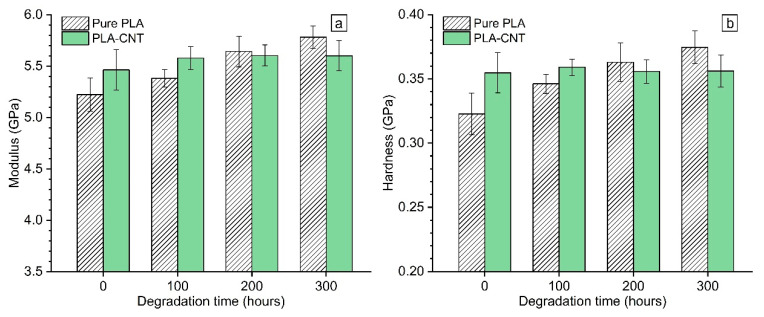
Evolution of (**a**) elastic modulus and (**b**) hardness of PLA and PLA-CNT during accelerated weathering degradation for up to 300 h.

**Table 1 biomimetics-05-00061-t001:** Thermal properties of PLA and its nanocomposite before and after degradation test.

Material	Time (h)	T_g_ (°C)	T_onset_ (°C)	T_P1_ (°C)	T_P2_ (°C)	X_c_ (%)
Pure PLA	0	61.3 ± 0.7	159.6 ± 0.3	166.9 ± 0.3	173.6 ± 0.1	38.9 ± 0.1
	100	63.4 ± 0.5	160.2 ± 0.5	167.7 ± 0.3	173.8 ± 0.2	42.4 ± 0.1
	200	61.3 ± 1.2	160.2 ± 0.8	168.0 ± 0.6	173.9 ± 0.3	43.6 ± 0.1
	300	60.3 ± 0.3	159.9 ± 0.2	169.1 ± 0.5	173.9 ± 0.3	45.2 ± 0.1
PLA-CNT	0	61.8 ± 1.1	160.7 ± 0.1	168.0 ± 0.2	172.8 ± 0.1	41.4 ± 0.1
	100	64.0 ± 0.6	161.7 ± 0.7	170.2 ± 0.5	175.2 ± 0.4	43.9 ± 0.1
	200	60.5 ± 0.9	161.2 ± 0.5	171.2 ± 0.7	176.1 ± 0.3	43.5 ± 0.1
	300	59.2 ± 1.3	161.0 ± 0.2	171.7 ± 0.6	176.3 ± 0.3	43.8 ± 0.1

Tg: the glass transition temperature.

**Table 2 biomimetics-05-00061-t002:** Thermal properties of PLA and its nanocomposite before and after degradation test.

Sample	No. of Voids/1.8 × 10^−2^ (mm^2^)	Average Void Size (µm)	Surface Porosity (%)
PLA-0 h	0	0	0
PLA-100 h	33 ± 5	6.976 × 10^−3^ ± 0.00082	0.015 ± 0.003
PLA-200 h	47 ± 8	1.0174 × 10^−2^ ± 0.00083	0.342 ± 0.003
PLA-300 h	88 ± 8	1.6294 × 10^−2^ ± 0.00037	1.031 ± 0.0012
PLA-CNT-0 h	1	1.0147 × 10^−2^	0.008
PLA-CNT-100 h	2 ± 1	6.468 × 10^−3^ ± 0.00079	0.01 ± 0.0028
PLA-CNT-200 h	10 ± 3	7.99 × 10^−3^ ± 0.00012	0.057 ± 0.001
PLA-CNT-300 h	22 ± 4	1.3116 × 10^−2^ ± 0.00036	0.207 ± 0.0016

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
