# Peer review of "Improved Weathering Performance of Poly(Lactic Acid) through Carbon Nanotubes Addition: Thermal, Microstructural, and Nanomechanical Analyses"

_biomimetics, 2020, doi:10.3390/biomimetics5040061_

Round 1
Reviewer 1 Report
This is a very interesting and well-written article on the usage of PLA for the development of sustainable materials.
In order to understand the interrelationship between the microstructure and degradation behaviour of poly (lactic acid) (PLA), single-walled carbon nanotubes (CNTs) were introduced into PLA as nucleating agents. The degradation behaviour of PLA-CNT nanocomposites was examined under accelerated weathering conditions with exposure to UV light, heat, and moisture. The degradation mechanism proceeded via the Norrish type II mechanism of carbonyl polyester.
CNTs were successfully found to increase the crystallinity of the PLA and modified the resultant morphology. Detailed characterization of the samples was carried out with an in-depth analysis of results and discussion.
Following minor corrections are needed:
-Can authors explain any specific reason for choosing PLA-CNT 1 wt.% ratio?
-Figure 1 should be in colour format
-A schematic for the nanocomposite preparation should be there
-The magnifications of SEM images are not visible, please clearly mention the magnification of each image
- Introduction should be strengthened by highlighting the importance of biopolymers by citing relevant articles such as Chemical Reviews 120 (17), 9304–9362 (2020); Progress in environmental-friendly polymer nanocomposite material from PLA: Synthesis, processing and applications, Vacuum 146, 655-663 (2017); Biomimetics 2020, 5(3), 42 (2020)
Author Response
Figures can be found in the attached file
- Can authors explain any specific reason for choosing PLA-CNT 1 wt.% ratio?
It has been known that the incorporation of low amount of CNTs into various polymer matrix would lead to significant improvements in physical properties. While reviewing the current literature, the authors have found a wide range of CNTs concentration reported in the literature. Desa et al. (Mat Desa et al. 2014) suggested that the concentration of CNTs should be between 1% to 5%. Basheer et al. (Basheer et al. 2020) reported that the addition of even a small amount (less than 5 wt.%) of nanofillers can increase the thermomechanical and electrical properties of the resulting polymer nanocomposites. Gonçalves et al. (Gonçalves et al. 2017) reported that the filler concentrations most often tested are between 0.1–2 wt.%.
In our case, based on the sample preparation methods which includes solution mixing and melt mixing, we found several studies that used 1 wt.% of nanofiller with the similar preparation methods (Yoon et al. 2010; Shokoohi and Arefazar 2009; Nature-Inspired Surface and Materials Research: (Adv. Mater. 45/2017) 2017). In addition, during the solution mixing process, we found that this concentration produced relatively uniform dispersion of CNTs within the polymer matrix. Higher filler concentrations would more likely lead to aggregation of the nanofiller. Therefore, 1 wt.% was selected for this study.
- Figure 1 should be in color format.
As per the reviewer’s suggestion, we have provided a color figure.
Figure 2. Typical load-displacement curve for the nanoindentation tests.
- A schematic for the nanocomposite preparation should be there.
The flowchart of the nanocomposite synthesis process is prepared and added to manuscript in the materials and methods section on page 3.
Figure 1. Fabrication process of the CNT-PLA thin films
- The magnifications of SEM images are not visible, please clearly mention the magnification of each image.
SEM images are revised as per the reviewer’s suggestions.
Figure 3. Scanning electron micrographs at 1000X and 2000X magnification of pure PLA (a-b), and PLA-CNT (c-d) after isothermal crystallization.
Figure 7. SEM results showing the surface morphology at 1000X magnification of (a-d) pure PLA, and (a’-d-) PLA-CNT after 300h of degradation.
Figure 8. SEM images at 10,000X magnification illustrating the changes in the surface morphology in terms of voids and holes during 300h of degradation; a-d: PLA spherulic lamellae; a’-d’: PLA-CNT spherulic lamellae.
- Introduction should be strengthened by highlighting the importance of biopolymers by citing relevant articles such as Chemical Reviews 120 (17), 9304–9362 (2020); Progress in environmental-friendly polymer nanocomposite material from PLA: Synthesis, processing and applications, Vacuum 146, 655-663 (2017); Biomimetics 2020, 5(3), 42 (2020).
Suggested articles have been included in the revised manuscript in the introduction section. Lines 33-35 as:
“Poly(lactic acid) (PLA), an aliphatic polyester, is one of the most promising biopolymers to replace conventional petroleum-based plastics [1]. The use of biopolymers has attracted significant attention because of increasing environmental concerns due to the waste accumulation and depletion of fossil resources (Ates et al. 2020; Dubey et al. 2017; Vidakis et al. 2020). PLA is a sustainably-sourced material, and can be derived from renewable resources such as corn, rice, or wheat. With the advantages of being a bio-compatible, hydrolysable, and biodegradable material, PLA has great potential for both packaging and biomedical applications [1,2].”
Publication bibliography
Ates, Burhan; Koytepe, Suleyman; Ulu, Ahmet; Gurses, Canbolat; Thakur, Vijay Kumar (2020): Chemistry, Structures, and Advanced Applications of Nanocomposites from Biorenewable Resources. In Chemical reviews 120 (17), pp. 9304–9362. DOI: 10.1021/acs.chemrev.9b00553.
Basheer, Bashida V.; George, Jinu Jacob; Siengchin, Suchart; Parameswaranpillai, Jyotishkumar (2020): Polymer grafted carbon nanotubes—Synthesis, properties, and applications: A review. In Nano-Structures & Nano-Objects 22, p. 100429. DOI: 10.1016/j.nanoso.2020.100429.
Dubey, Satya P.; Thakur, Vijay K.; Krishnaswamy, Suryanarayanan; Abhyankar, Hrushikesh A.; Marchante, Veronica; Brighton, James L. (2017): Progress in environmental-friendly polymer nanocomposite material from PLA: Synthesis, processing and applications. In Vacuum 146, pp. 655–663. DOI: 10.1016/j.vacuum.2017.07.009.
Gonçalves, Carolina; Gonçalves, Inês C.; Magalhães, Fernão D.; Pinto, Artur M. (2017): Poly(lactic acid) Composites Containing Carbon-Based Nanomaterials: A Review. In Polymers 9 (7). DOI: 10.3390/polym9070269.
Mat Desa, M. S. Z.; Hassan, A.; Arsad, A.; Mohammad, N. N. B. (2014): Mechanical properties of poly(lactic acid)/multiwalled carbon nanotubes nanocomposites. In Materials Research Innovations 18 (sup6), S6-14-S6-17. DOI: 10.1179/1432891714Z.000000000924.
Nature-Inspired Surface and Materials Research: (Adv. Mater. 45/2017) (2017). In Adv. Mater. 29 (45).
Shokoohi, Shirin; Arefazar, Ahmad (2009): A review on ternary immiscible polymer blends: morphology and effective parameters. In Polym. Adv. Technol. 20 (5), pp. 433–447. DOI: 10.1002/pat.1310.
Vidakis, Nectarios; Petousis, Markos; Velidakis, Emmanouel; Liebscher, Marco; Tzounis, Lazaros (2020): Three-Dimensional Printed Antimicrobial Objects of Polylactic Acid (PLA)-Silver Nanoparticle Nanocomposite Filaments Produced by an In-Situ Reduction Reactive Melt Mixing Process. In Biomimetics (Basel, Switzerland) 5 (3). DOI: 10.3390/biomimetics5030042.
Yoon, Jin Tae; Lee, Sang Cheol; Jeong, Young Gyu (2010): Effects of grafted chain length on mechanical and electrical properties of nanocomposites containing polylactide-grafted carbon nanotubes. In Composites Science and Technology 70 (5), pp. 776–782. DOI: 10.1016/j.compscitech.2010.01.011.

Reviewer 2 Report
The nucleating effects of CNT on PLA and the effects of CNT on degradation behavior of PLA were investigated. Obtained results are interesting but manuscript requires revisions considering the following remarks.
(1) The molecular weight change of PLA samples during accelerated weathering test affects the thermal and mechanical properties and therefore it is recommended to be measured.
(2) The crystalline modification (delta or alpha) should be determined by WAXD not DSC.
(3) The intensities of FTIR spectra are recommended to be equalized. The gradual increase of intensity with degradation time in Figure 5 seems to mislead the results. For example, the increase in intensity at 1757 cm-1 of PLA samples. The number of carbonyl groups do not change by Norrish Type II degradation and hydrolytic degradation.
(4) Please specify the temperature for accelerating weathering test. Increased temperature and water as a plasticizer should induce the stabilized chain packing in the amorphous region, which effect is stronger compared to that of lowered molecular weight at the initial stage. The result is very similar to pure hydrolytic degradation reported in J. Appl. Polym. Sci., 2000, 77, 1452-1464. The reason for the increased Tg stated in lines 254 and 255 is recommended to be reconsidered.
(5) The nucleating effects of carbon nanotubes are reported in Polymer 2007, 48, 4213-4225 and detailed study on the effect of UV-irradiation of PLLA on hydrolytic degradation is reported in J. Appl. Polym. Sci., 2012, 125, 2394-2406. The obtained results are recommended to be compared with the reported results.
(6) The increase in Modulus and Hardness of PLA films during degradation indicates that the effects of pore formation in the surface is slight compared to the increased crystallinity. This is recommended to be stated in the manuscript.
(7) It might better to start a new paragraph with "Figure 7 shows…" on page 9.
Author Response
The nucleating effects of CNT on PLA and the effects of CNT on degradation behavior of PLA were investigated. Obtained results are interesting but manuscript requires revisions considering the following remarks.
(1) The molecular weight change of PLA samples during accelerated weathering test affects the thermal and mechanical properties and therefore it is recommended to be measured.
We agree that adding the molecular weight (MW) analysis would certainly be of interest for future work. Currently, we do not have the capability to investigate the effect of polymer degradation on molecular weight due to the lack of proper equipment. However, the authors believe that the combination of microscopy, thermal, and nanoindentation analyses are sufficient to support the conclusions drawn in the manuscript. In addition, the degradation was demonstrated by reduction in thermal stability which, in turn, is related to a decrease in MW.
(2) The crystalline modification (delta or alpha) should be determined by WAXD not DSC.
The author agreed that XRD is a useful method to characterize crystal structure, grain size, or crystallinity. Both XRD and DSC are very convenient in the crystallinity studies and their information are usually complementary. However, in this study, we wanted to focus more on the rate of crystallization, phase change, and the re-crystallization phenomena from α’ to α-crystals. Currently, we do not have XRD results for these samples, but previous studies confirmed that α’ to α-phases in PLA can be studied using DSC. Our DSC results and subsequent discussion are being made by comparison with the following articles: (Macromolecules. 44 (2011) 6496–6502; Macromol. 2007, 40, 26, 9463–9469; Polym. Adv. Technol., 27: 844– 859).
(3) The intensities of FTIR spectra are recommended to be equalized. The gradual increase of intensity with degradation time in Figure 5 seems to mislead the results. For example, the increase in intensity at 1757 cm-1 of PLA samples. The number of carbonyl groups do not change by Norrish Type II degradation and hydrolytic degradation.
Since there is some debate on the significance and interpretation of the FTIR results, they were removed from the manuscript. We can still make significant conclusions regarding the improved weathering properties of PLA composites without these results using thermal and nano-mechanical analysis. The FTIR results with the equalized intensities can be included as supplementary data, if desired.
(4) Please specify the temperature for accelerating weathering test. Increased temperature and water as a plasticizer should induce the stabilized chain packing in the amorphous region, which effect is stronger compared to that of lowered molecular weight at the initial stage. The result is very similar to pure hydrolytic degradation reported in J. Appl. Polym. Sci., 2000, 77, 1452-1464. The reason for the increased Tg stated in lines 254 and 255 is recommended to be reconsidered.
The temperature for weathering test is mentioned in section 2.3, lines 110-112 as:
“...Fluorescent lamps (UVA-340) with 0.89W/m2 (at 340 nm) irradiance were used with cycles of 8h UV exposure irradiation at 60 °C followed by 4h of dark condensation at 50 °C…”.
After reviewing the recommended study, we agreed to change the statement in section 3.2.1, lines 266-272 in the revised manuscript as:
“…The tendency of increasing Tg at the beginning of the degradation process was previously reported [ J. Appl. Polym. Sci., 2000, 77, 1452-1464]. This is likely due to stabilized chain packing in the amorphous region by low temperature annealing in the presence of water molecules which are acting as a plasticizer [J. Appl. Polym. Sci., 2000, 77, 1452-1464]. Hence, more energy is required to activate the glass-rubber transition, resulting in an increase in Tg. This increase in Tg leads to embrittlement which is one of the deleterious effects of degradation. However, as the degradation proceeded, Tg gradually decreased…”.
(5) The nucleating effects of carbon nanotubes are reported in Polymer 2007, 48, 4213-4225 and detailed study on the effect of UV-irradiation of PLLA on hydrolytic degradation is reported in J. Appl. Polym. Sci., 2012, 125, 2394-2406. The obtained results are recommended to be compared with the reported results.
Thank you for suggesting these references. We found many in-depth discussions in these articles that are closely related to our study:
Results from our study indicated the nucleating effect of CNTs where they acted as nucleating agents to facilitate overall crystallization of PLA. We found that similar result was reported in Polymer 2007, 48, 4213-4225. Hence, the authors have added the comparison in section 3.1.2, lines 223-234 in the revised manuscript as:
” However, the peak temperature and magnitude of these two crystalline structures slightly changed after adding CNTs. Pure PLA crystals started to melt at 159.6 °C, and the first peak was observed at 166.9 °C, while majority of crystallites was formed at the second peak at 173.6 °C. With the reinforcement of CNT, onset melting temperature increased to 160.7 °C, while the peak melting temperatures of bimodal peaks changed to 168.0 °C and 172.8 °C, respectively. Interestingly, the magnitude of first peak increased significantly with the inclusion of CNTs which indicates that PLA was crystallizing at much lower crystallization temperature. It happened due to the presence of CNTs where they acted as nucleating agents to accelerate overall crystallization [29]. Similar results have been observed in the past study [Polymer 2007, 48, 4213-4225].”
From J. Appl. Polym. Sci., 2012, 125, 2394-2406, there are couple of points that the authors found relevant to our study and used them accordingly as references to compare with our results. Please find them below as points and where the discussions are added to the revised manuscript:
- Crystallinity change: crystallinity initially increased at the beginning of degradation process, in our case, under photo and hydrolytic degradation simultaneously. This phenomenon which was previously reported in the suggested publication was confirmed with our DSC results. Accordingly, the related discussions are added to the revised manuscript in section 3.2.1, lines 282-289 as:
“The crystallinity increased monotonically from 38.9 % at 0 h to 42.35 % at 100h, 43.6 % at 200 h, and 45.15 % at 300 h (Table 1). Similar result was found in Tsuji et al. [J. Appl. Polym. Sci., 2012, 125, 2394-2406] where the relative crystallinity of amorphous PLA film increased rapidly in the first 12 hours due to the crystallization on immersion in phosphate-buffered solution. The increase in crystallinity with degradation is a well-known phenomenon: during degradation, chain scission occurs providing the polymer chains with extra mobility to rearrange into larger crystal structures [30]. Also, secondary crystallization can proceed below Tg over the time scale of months and years, so the PLA may have simply continued to crystallize over the extended period of time [40].”
- Crystallization is more photo degradation resist than amorphous.
The following has been included in the revised manuscript in section 3.2.2, lines 349-356 as:
“Expectedly, these micro voids first appeared in the amorphous regions in the polymer matrix since it has lower resistance to degradation than crystalline domains. It has previously been reported that both photolysis and hydrolytic degradation would start from the amorphous regions first [13, J. Appl. Polym. Sci., 2012, 125, 2394-2406]. Tsuji et al confirmed that the crystalline region has higher photodegradation resistance than amorphous region while monitoring molecular weight distribution of UV treated and untreated samples [J. Appl. Polym. Sci., 2012, 125, 2394-2406]. In another study, it was reported that the chain cleavage reaction during the hydrolytic degradation of PLA would proceed preferentially in amorphous regions [13].
(6) The increase in Modulus and Hardness of PLA films during degradation indicates that the effects of pore formation in the surface is slight compared to the increased crystallinity. This is recommended to be stated in the manuscript.
We appreciate the reviewer for raising this important point. Volume under the test (VUT) in nanoindentation includes part of bulk material which is under stress and acts against the penetration (Nikaeen et al. 2019). A large VUT represents properties of the bulk material regarding the inclusion of nano/micro defects existing in the structure of the nanocomposite. These defects could include air pockets, CNF agglomerations, microcracks, pores and any other defects created during fabrication/degradation. Given the sizes of the pores generated by the degradation from the SEM micrographs from figure 8, one can conclude that a 10 µm deep nanoindentation is deep enough to include all microstructural characteristics of the nanocomposite such as pores and CNFs. Therefore, with pore formation acting against the mechanical integrity of nanocomposites and increased crystallinity acting in its favor, the increase in modulus and hardness of PLA films upon degradation indicates that the effect of pore formation in the surface is slight compared to the increased crystallinity.
The aforementioned statement is moved to the section 3.2.3, lines 419-421.
(7) It might better to start a new paragraph with "Figure 7 shows…" on page 9.
We agreed. Manuscript has been revised accordingly.
Publication bibliography
Nikaeen, Peyman; Depan, Dilip; Khattab, Ahmed (2019): Surface Mechanical Characterization of Carbon Nanofiber Reinforced Low-Density Polyethylene by Nanoindentation and Comparison with Bulk Properties. In Nanomaterials (Basel, Switzerland) 9 (10). DOI: 10.3390/nano9101357.

Round 2
Reviewer 2 Report
The manuscript has been well-revised. The manuscript can be published in the present form.